# Risk of Coronavirus Disease 2019 (COVID-19) among those up-to-date and not up-to-date on COVID-19 vaccination by US CDC criteria

**Nabin K. Shrestha** [1]*, **Patrick C. Burke**[2], **Amy S. Nowacki**[3], **Steven M. Gordon**[1]

**1** Department of Infectious Diseases, Cleveland Clinic, Cleveland, Ohio, United States of America,
**2** Infection Prevention, Cleveland Clinic, Cleveland, Ohio, United States of America, **3** Quantitative Health Sciences, Cleveland Clinic, Cleveland, Ohio, United States of America

* shrestn@ccf.org

**Data Availability Statement:** Data are available from https://osf.io/bufwc/.

**Funding:** The authors received no specific funding for this work.

## Abstract

### Background

The CDC recently defined being "up-to-date" on COVID-19 vaccination as having received at least one dose of a COVID-19 bivalent vaccine. The purpose of this study was to compare the risk of COVID-19 among those "up-to-date" and "not up-to-date".

### Methods

Employees of Cleveland Clinic in Ohio, USA, in employment when the COVID-19 bivalent vaccine first became available, and still employed when the XBB lineages became dominant, were included. Cumulative incidence of COVID-19 since the XBB lineages became dominant was compared across the"up-to-date" and "not up-to-date" states, by treating COVID-19 bivalent vaccination as a time-dependent covariate whose value changed on receipt of the vaccine. Risk of COVID-19 by vaccination status was also evaluated using multivariable Cox proportional hazards regression adjusting for propensity to get tested for COVID-19, age, sex, most recent prior SARS-CoV-2 infection, and number of prior vaccine doses.

### Results

COVID-19 occurred in 1475 (3%) of 48 344 employees during the 100-day study period. The cumulative incidence of COVID-19 was lower in the "not up-to-date" than the "up-to-date" state. On multivariable analysis, being "up-to-date" was not associated with lower risk of COVID-19 (HR, 1.05; 95% C.I., 0.88–1.25; P-value, 0.58). Results were very similar when those 65 years and older were only considered "up-to-date" after 2 doses of the bivalent vaccine.

### Conclusions

Since the XBB lineages became dominant, adults "up-to-date" on COVID-19 vaccination by the CDC definition do not have a lower risk of COVID-19 than those "not up-to-date", bringing into question the value of this risk classification definition.

**Competing interests:** The authors have declared that no competing interests exist.

# Introduction

In April 2023 the Centers for Disease Control and Prevention (CDC) updated its guidance on Coronavirus Disease 2019 (COVID-19) vaccination to consider all individuals above the age of 6 to be "up-to-date" with COVID-19 vaccination only if they had received at least one dose of a COVID-19 bivalent vaccine [1]. By this definition, those who had not received a single dose of a COVID-19 bivalent vaccine would be considered not "up-to-date". A recent study was unable to find the bivalent vaccine to be effective while the XBB lineages were the dominant circulating strains [2]. Given this lack of effectiveness of the bivalent vaccine against the XBB lineages, it was reasonable to question whether individuals "up-to-date" with a vaccine of questionable effectiveness were protected against COVID-19 compared to those not "up-to-date", when the XBB lineages were the predominant circulating strains.

The purpose of this study was to evaluate whether individuals who were "up-to-date" on COVID-19 vaccination had a lower risk of COVID-19 than those not "up-to-date".

# Methods

## Study design

This was a retrospective cohort study conducted at the Cleveland Clinic Health System (CCHS) in the United States.

## Patient consent statement

The study was approved by the Cleveland Clinic Institutional Review Board as exempt research (IRB no. 22–917). A waiver of informed consent and waiver of HIPAA authorization were approved to allow the research team access to the required data.

## Setting

Since the arrival of the COVID-19 pandemic at Cleveland Clinic in March 2020, employee access to testing has been a priority. Voluntary vaccination for COVID-19 began on 16 December 2020, and the monovalent mRNA vaccine as a booster became available to employees on 5 October 2021. The COVID-19 bivalent mRNA vaccine began to be offered to employees on 12 September 2022. The study start date was 29 January 2023, the date the XBB lineages first became the dominant circulating strains in Ohio.

## Participants

CCHS employees in employment at any location in Ohio when the COVID-19 bivalent vaccine first became available (12 September 2022), and who were still employed when the XBB lineages became dominant (29 January 2023), were included in the study. Those for whom age and sex were not available were excluded.

## Variables

Covariates collected were age, sex, job location, prior SARS-CoV-2 infection, and number of prior vaccine doses, as described in our earlier studies [2–5]. Institutional data governance rules related to employee data limited our ability to supplement our dataset with additional clinical variables. Vaccination status was "up-to-date" or "not up-to-date" based on whether or not a person had received at least one dose of a COVID-19 bivalent vaccine. For each subject, the propensity to get tested for COVID-19 was defined as the number of COVID-19 nucleic acid amplification tests (NAATs) done divided by the number of years of employment at

CCHS during the pandemic. The pandemic phase during which a subject had his or her last prior episode of COVID-19 was also collected as a variable, based on which variant/lineages accounted for more than 50% of infections in Ohio at the time [6].

### Outcome

The study outcome was time to COVID-19, the latter defined as a positive NAAT for Severe Acute Respiratory Syndrome Coronavirus 2 (SARS-CoV-2) any time after the study start date. Outcomes were followed until 10 May 2023, allowing for evaluation of outcomes up to 100 days from the study start date. Data were accessed on 11 May 2023. Authors had access to information that could identify individual participants. However all data were de-identified before analysis.

### Statistical analysis

A Simon-Makuch hazard plot [7] was created to compare the cumulative incidence of COVID-19 in the "up-to-date" and "not up-to-date" states, by treating vaccination status as a time-dependent covariate. A subject's vaccination status was "not up-to-date" before receipt of a COVID-19 bivalent vaccine, and "up-to-date" after receipt of the vaccine. Subjects whose employment was terminated during the study period before they had COVID-19 were censored on the date of termination. Curves for the "not up-to-date" state were based on data while the vaccination status was "not up-to-date". Curves for the "up-to-date" state were based on data from the date the vaccination status changed to "up-to-date".

Multivariable Cox proportional hazards regression models were fit to examine the association of various variables with time to COVID-19. Vaccination status was included as a time-dependent covariate [8]. The possibility of multicollinearity in the models was evaluated using variance inflation factors. The proportional hazards assumption was checked using log(-log (survival)) vs. time plots. The hazard ratio for vaccination status provided a comparison of risk of COVID-19 based on vaccination status after adjusting for all other covariates.

The analysis was performed by N. K. S. and A.S.N. using the *survival* package and R version 4.2.2 (R Foundation for Statistical Computing) [8–10].

## Results

Of 48 344 included subjects, 1445 (3%) were censored during the study because of termination of employment. By the end of the study, 12 841 (27%) were "up-to-date" on COVID-19 vaccination, of whom 11 187 (87%) received the Pfizer vaccine and 1654 (13%) received the Moderna vaccine. Those "up-to-date" by the end of the study included 37 of the 6174 who had previously been unvaccinated at the start of the study. A total of 1475 employees (3%) acquired COVID-19 during the 100 days of the study.

### Baseline characteristics

Table 1 shows the characteristics of subjects included in the study. Notably, this was a relatively young population, with a mean age of 43 years. Among these, 22 407 (46%) had previously had a documented episode of COVID-19 and 16 262 (34%) had previously had an Omicron variant infection. 42 160 subjects (87%) had previously received at least one dose of vaccine and 44 432 (92%) had been previously exposed to SARS-CoV-2 by infection or vaccination. Altogether, 36 490 subjects (76%) were tested for COVID-19 by a NAAT at least once while employed at Cleveland Clinic. The propensity for COVID-19 testing ranged from 0 to 63.5 per year, with a median of 0.64 and interquartile range spanning 0.32 to 1.27 per year.

**Table 1. Baseline characteristics of 48 344 employees of cleveland clinic in Ohio.**

| Characteristics | Overall |
|---|---|
| Age in years, mean (SD) | 42.5 (13.2) |
| Sex | |
| Female | 35 953(74.4) |
| Male | 12 381 (25.6) |
| Location | |
| Cleveland Clinic Main | 19 445 (40.2) |
| Cleveland area regional hospitals | 11 355 (23.5) |
| Ambulatory centers | 8443 (17.5) |
| Cleveland Clinic Akron | 3996 (8.3) |
| Administrative centers | 3971 (8.2) |
| Cleveland Clinic Medina | 1124 (2.3) |
| Prior documented COVID-19[a] | 22 407 (46.4) |
| Prior Omicron variant infection[b] | 16 262 (33.6) |
| Most recent prior SARS-CoV-2 infection | |
| No documented infection | 25 927 (53.6) |
| During the pre-Omicron phase | 6145 (12.7) |
| During the Omicron BA.1/BA.2 dominant phase | 8876 (18.4) |
| During the Omicron BA.4/BA.5 dominant phase | 5943 (12.3) |
| During the Omicron BQ dominant phase | 1443 (3.0) |
| Number of prior vaccine doses | |
| 0 | 6174 (12.8) |
| 1 or 2 | 16 301 (33.7) |
| 3 | 14 731 (30.5) |
| >3 | 11 128 (23.0) |
| Days since most recent infection[c], mean (SD) | 365 (238) |
| Days since most recent vaccine[d], mean (SD) | 370 (214) |
| Days since proximate SARS-CoV-2 exposure[e], mean (SD) | 306 (200) |

Data are presented as no. (%) unless otherwise indicated.

Abbreviation: SD, standard deviation; COVID-19, Coronavirus Disease 2019; SARS-CoV-2, Severe Acute Respiratory Syndrome Coronavirus-2.

[a]Defined by a positive SARS-CoV-2 nucleic acid amplification test.

[b]Defined by a positive SARS-CoV-2 nucleic acid amplification test during a time when Omicron variants were the predominant strains in Ohio.

[c]Among those with prior documented COVID-19.

[d]Among those who received at least one dose of a COVID-19 vaccine.

[e]Among those with prior documented COVID-19 or who received at least one dose of a COVID-19 vaccine.

## Risk of COVID-19 based on vaccination status and prior infection

The risk of COVID-19 was lower in the "not up-to-date" state than in the "up-to-date" state, with respect to COVID-19 vaccination (Fig 1). When stratified by tertiles of propensity to get tested for COVID-19, the "up-to-date" state was not associated with a lower risk of COVID-19 than the "not up-to-date" state in any tertile (Fig 2).

In contrast, consideration of prior infection provided a more accurate classification for risk of COVID-19. Those whose last prior SARS-CoV-2 infection occurred during the Omicron

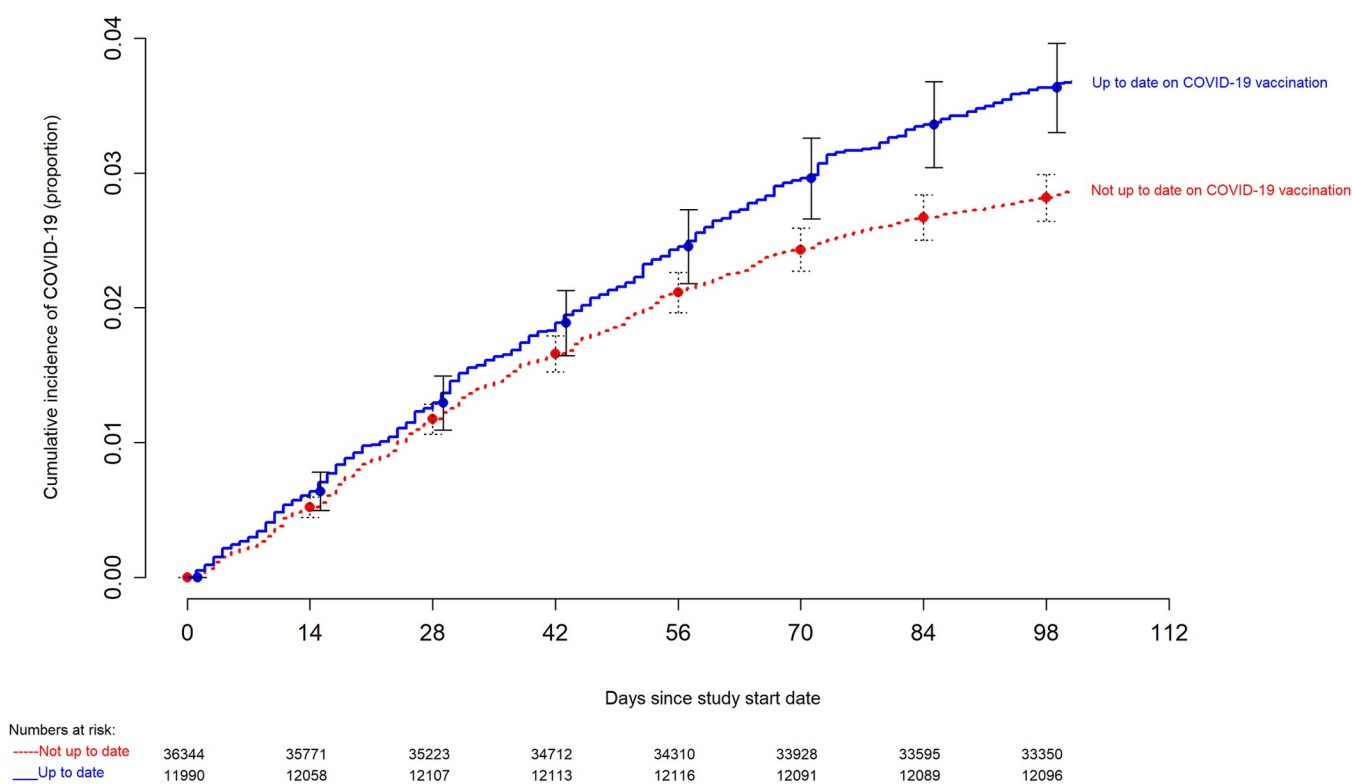

**Fig 1. Simon-Makuch hazard plot comparing the cumulative incidence of COVID-19 in the "up-to-date" and "not up-to-date" states with respect to COVID-19 vaccination.** Day zero was 29 January 2023, the day the XBB lineages of the Omicron variant became the dominant strains in Ohio. Point estimates and 95% confidence intervals are jittered along the x-axis to improve visibility.

BQ or BA.4/BA.5 dominant phases had a substantially and significantly lower risk of COVID-19 than those whose last SARS-CoV-2 infection occurred in the early Omicron or pre-Omicron periods or those not previously known to have had COVID-19 (Fig 3). When stratified by most recent prior SARS-CoV-2 infection, there was no difference in risk of COVID-19 between the "up-to-date" and "not up-to-date" states within each most recent prior infection group, except for those not previously known to be infected, among whom the "up-to-date" state was associated with a higher risk of COVID-19 than the "not up-to-date" state (Fig 4).

## Multivariable analysis

In a multivariable Cox proportional hazards regression model, adjusted for propensity to get tested for COVID-19, age, sex, phase of most recent SARS-CoV-2 infection, and number of prior vaccine doses, being "up-to-date" on COVID-19 vaccination was not associated with a lower risk of COVID-19 than not being "up-to-date" (HR, 1.05; 95% C.I., 0.88–1.25; P-value, 0.58). Having had their last prior episode of COVID-19 while lineages of the Omicron variant were dominant was protective against COVID-19 with higher protection the more recent the infection. A higher number of prior vaccine doses was associated with higher risk of COVID-19. Point estimates of hazard ratios and associated 95% confidence intervals for the variables included in the unadjusted and adjusted Cox proportional hazards regression models are shown in Table 2. If number of prior vaccine doses was not adjusted for, being "up-to-date" on COVID-19 vaccination was associated with a higher risk of COVID-19 that not being "up-to-

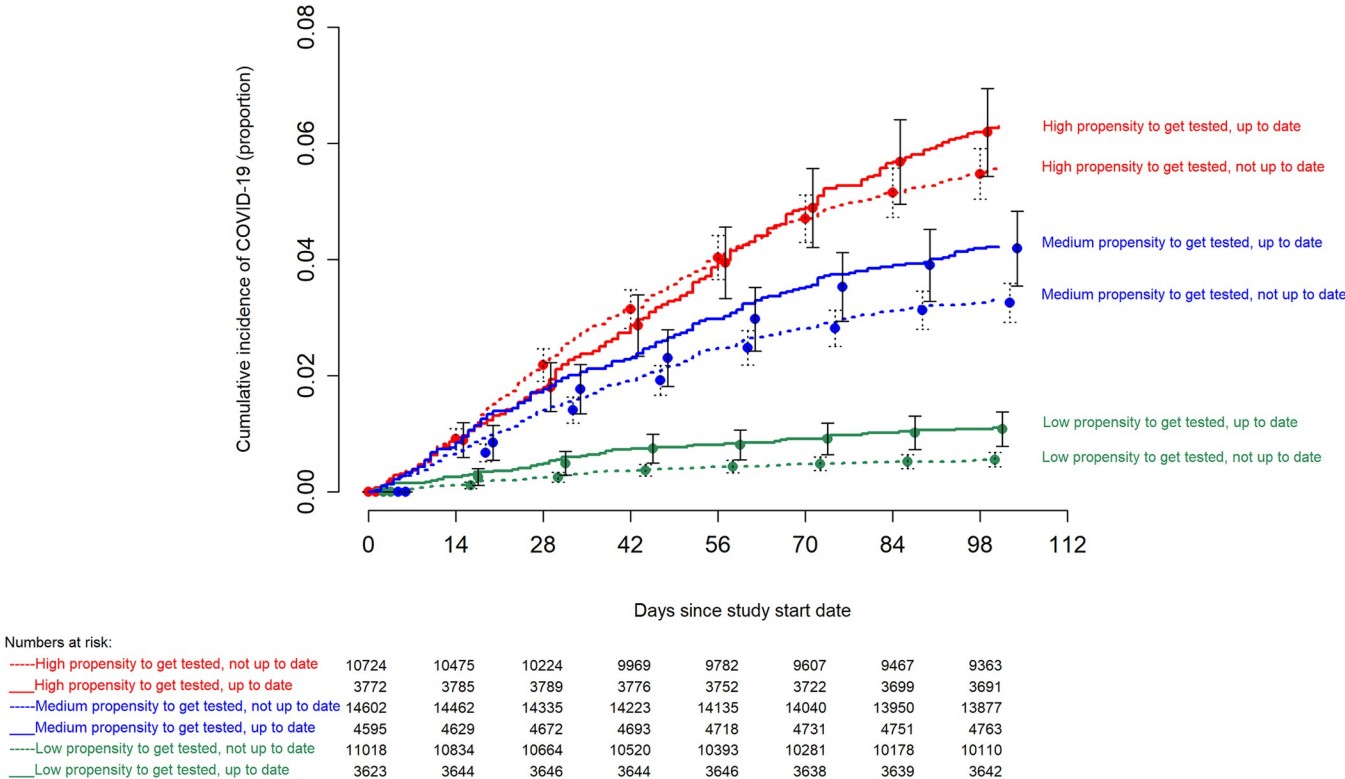

**Fig 2. Simon-Makuch hazard plot comparing the cumulative incidence of COVID-19 in the "up-to-date" and "not up-to-date" states with respect to COVID-19 vaccination, stratified by tertiles of propensity to get tested for COVID-19.** Day zero was 29 January 2023, the day the XBB lineages of the Omicron variant became the dominant strains in Ohio. Point estimates and 95% confidence intervals are jittered along the x-axis to improve visibility. Solid lines represent the "up-to-date" states while dashed lines represent the "not up-to-date" states.

date" (HR, 1.30; 95% C.I., 1.16–1.45; *P*-value, <0.001). Results were very similar if those 65 years and older were only considered "up-to-date" after receipt of at least 2 doses of a bivalent vaccine (Table 1). Results were also very similar if an individual was only considered "up-to-date" 7 days after receipt of a bivalent vaccine (Table 2).

## Discussion

This study found that being "up-to-date" on COVID-19 vaccination, using the current CDC definition, was not associated with a lower risk of COVID-19 than not being "up-to-date", while the XBB lineages were the dominant circulating strains of SARS-CoV-2, after adjusting for important factors that might influence risk of COVID-19.

The strengths of our study include its large sample size, and its conduct in a healthcare system that devoted resources to have an accurate accounting of who had COVID-19, when COVID-19 was diagnosed, who received a COVID-19 vaccine, and when. The study methodology, treating vaccination status as a time-dependent covariate, allowed for determination of vaccine effectiveness in real time. Adjusting for the propensity to get tested for COVID-19 should have mitigated against bias that might arise from infections being more likely to be detected in individuals who tested more frequently. Furthermore, those who received the COVID-19 bivalent vaccine did so with an expectation that it provided protection against COVID-19. So, if there was a difference in likelihood of getting tested before and after

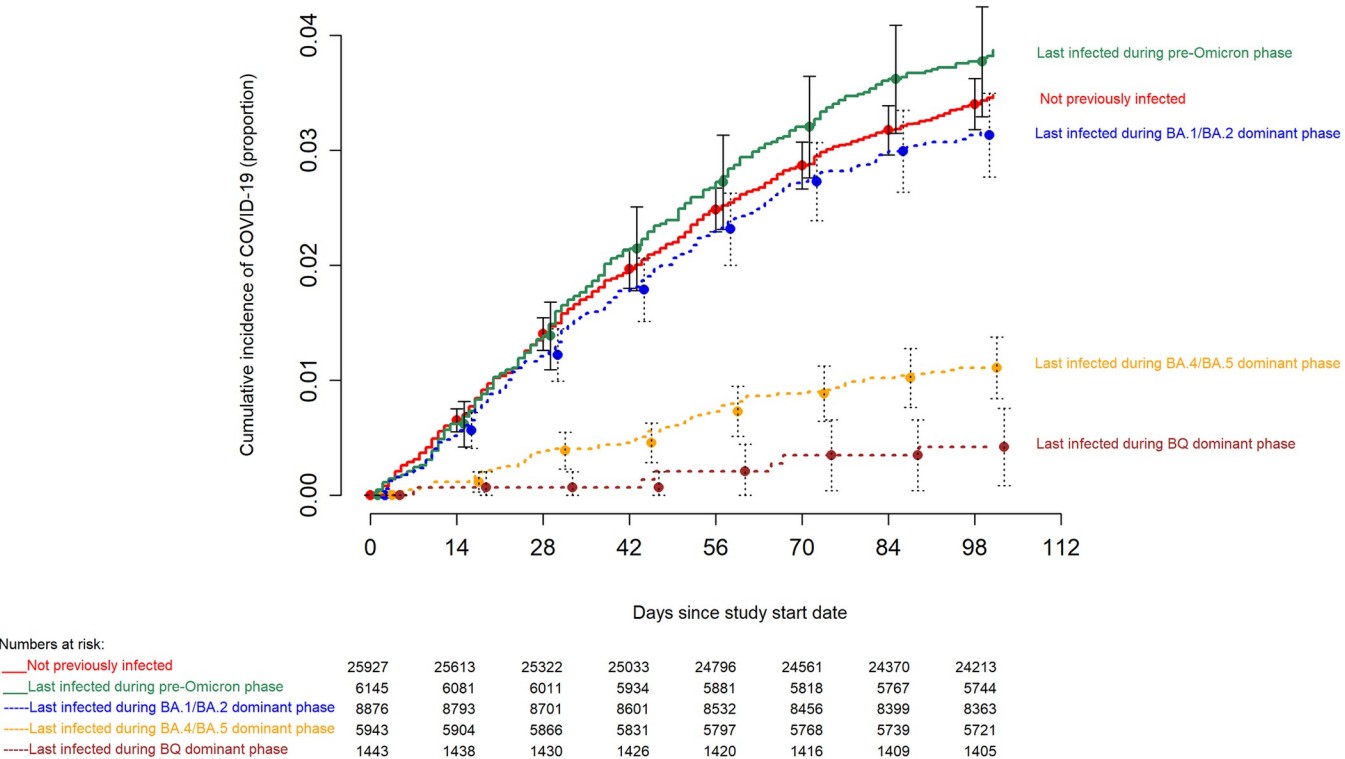

**Fig 3. Hazard plot comparing the cumulative incidence of COVID-19 stratified by the phase during which the last prior episode of COVID-19 occurred.** Day zero was 29 January 2023, the day the XBB lineages of the Omicron variant became the dominant strains in Ohio. Point estimates and 95% confidence intervals are jittered along the x-axis to improve visibility. Solid lines represent those last infected before the Omicron variants became dominant or those not previously known to have had COVID-19, while dashed lines represent those last infected while the Omicron variants were dominant.

receiving the COVID-19 bivalent vaccine, one would expect relatively less testing in the "up-to-date" state than in the "not up-to-date" state, having the effect of undercounting infections in the "up-to-date" state, thereby falsely overestimating the beneficial effect of being "up-to-date".

Our study was limited to examination of all detected infections. We were unable to distinguish between symptomatic and asymptomatic infections, and the rarity of severe illnesses precluded studying whether being "up-to-date" decreased severity of illness. Information on prior COVID-19 would have been incomplete, as many asymptomatic and mildly symptomatic infections would have been missed. Infections diagnosed by in-home testing without follow-up PCR testing would have been missed, but there is no reason to think these would have been missed in the "up-to-date" and "not up-to-date" states at rates different enough to change the results significantly. There may have been unconsidered variables that might have influenced the results. Lastly, our study was done in a healthcare population, and included no children and few elderly subjects, and the majority of study subjects would not have been immunocompromised.

Specific prior infections, prior vaccinations, time since proximate exposure to SARS-CoV-2 by prior infection or vaccination, number of prior vaccine doses, and interval between vaccines, all possibly contribute to determining an individual's risk of acquiring COVID-19. It is difficult to account for the effects of all these factors in an individual. What this study clearly shows is that a simplistic categorization of being or not being "up-to-date" on COVID-19 bivalent vaccination did not provide an accurate classification of risk of acquiring COVID-19 while the XBB lineages were the dominant circulating strains. The questionable effectiveness

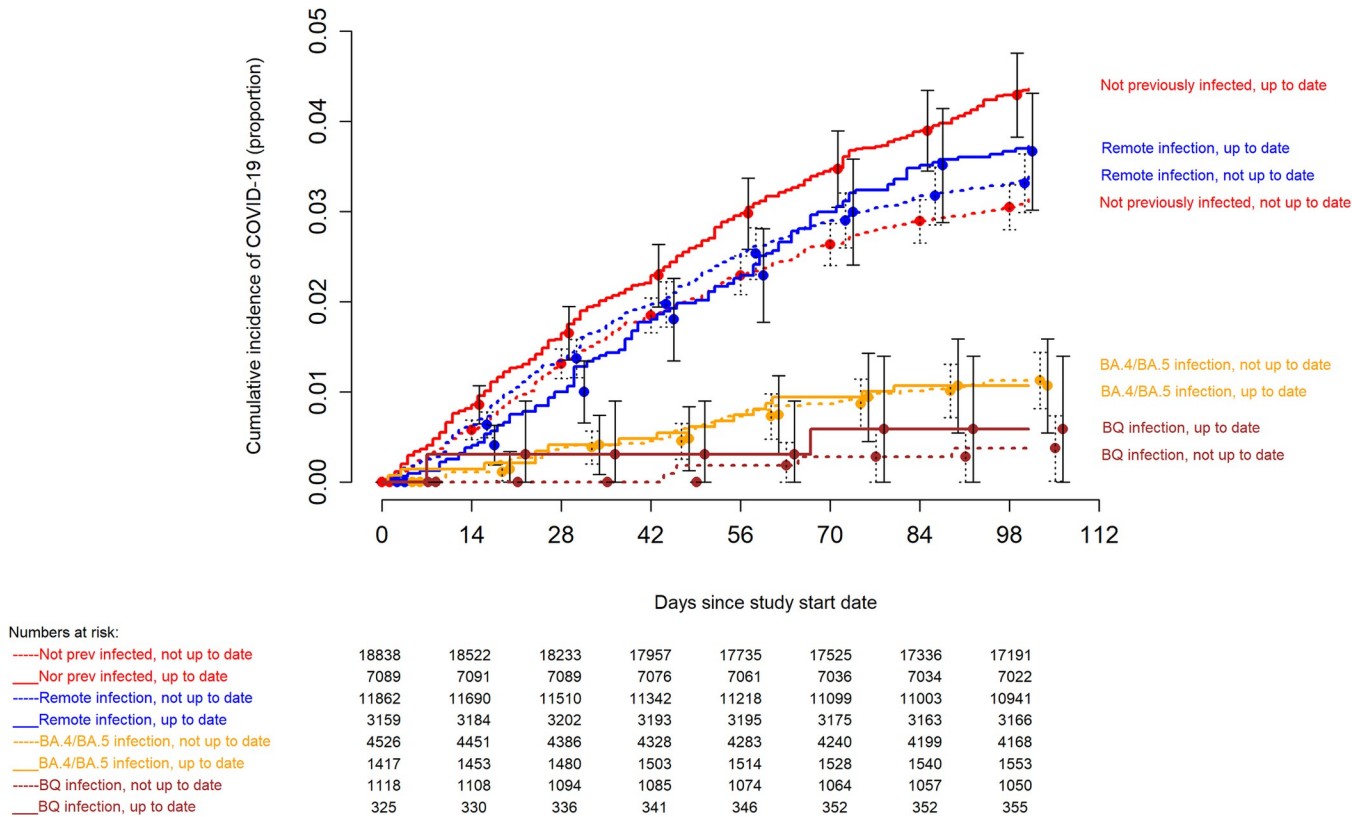

**Fig 4. Simon-Makuch hazard plot comparing the cumulative incidence of COVID-19 in the "up-to-date" and "not up-to-date" states with respect to COVID-19 vaccination, stratified by the phase during which the last prior episode of COVID-19 occurred.** Day zero was 29 January 2023, the day the XBB lineages of the Omicron variant became the dominant strains in Ohio. Point estimates and 95% confidence intervals are jittered along the x-axis to improve visibility. "Remote infection" includes infections during the pre-Omicron and Omicron BA.1/BA.2 dominant periods, i.e more than 218 days before the study start date. Solid lines represent the "up-to-date" states while dashed lines represent the "not up-to-date" states.

of the vaccine, as has been previously demonstrated [2], is likely a significant contributor to the lack of accuracy of risk classification based on receipt of the vaccine.

This study's findings question the wisdom of promoting the idea that every person needs to be "up-to-date" on COVID-19 vaccination, as currently defined, at this time. It is often stated that the primary purpose of vaccination is to prevent severe COVID-19 and death. We certainly agree with this, but it should be pointed out that there has not yet been a single study that has shown that the COVID-19 bivalent vaccine protects against severe disease or death caused by the XBB lineages of the Omicron variant. At least one prior study has failed to find a protective effect of the bivalent vaccine against the XBB lineages of SARS-CoV-2 [2]. Additionally, the finding of higher risk of COVID-19 with higher number of prior vaccine doses in this and our prior study [2], should be taken into consideration. People may still choose to get the vaccine, but an assumption that the vaccine protects against severe disease and death is not reason enough to unconditionally push a vaccine of questionable effectiveness to all adults.

In conclusion, this study found that being "up-to-date" on COVID-19 vaccination by the CDC definition was not associated with a lower risk of COVID-19 than not being "up-to-date". This study highlights the challenges of counting on protection from a vaccine when the effectiveness of the vaccine decreases over time as new variants emerge that are antigenically very different from those used to develop the vaccine. It also demonstrates the folly of risk classification based solely on receipt of a vaccine of questionable effectiveness while ignoring protection provided by prior infection.

**Table 2. Unadjusted and adjusted associations With time to COVID-19.**

| Variables | Unadjusted HR (95% CI) | P | Adjusted HR (95% CI)[a] | P |
|---|---|---|---|---|
| Vaccination status "up-to-date"[b] | 1.28 (1.15–1.43) | <0.001 | 1.05 (0.88–1.25) | 0.58 |
| Propensity to get tested for COVID-19[c] | 1.07 (1.06–1.09) | <0.001 | 1.09 (1.08–1.10) | <0.001 |
| Age | 1.001 (0.997–1.005) | 0.56 | 0.997 (0.993–1.001) | 0.09 |
| Male sex | 0.83 (0.74–0.94) | 0.004 | 0.80 (0.70–0.90) | <0.001 |
| Most recent prior SARS-CoV-2 infection[d] | | | | |
| During Pre-Omicron phase | 1.11 (0.96–1.29) | 0.14 | 1.07 (0.92–1.24) | 0.38 |
| During Omicron BA.1/BA.2 dominant phase | 0.91 (0.80–1.04) | 0.18 | 0.83 (0.72–0.95) | 0.006 |
| During Omicron BA.4/BA.5 dominant phase | 0.31 (0.24–0.40) | <0.001 | 0.27 (0.21–0.34) | <0.001 |
| During Omicron BQ dominant phase | 0.12 (0.05–0.27) | <0.001 | 0.09 (0.04–0.21) | <0.001 |
| Number of prior vaccine doses[e] | | | | |
| 1 or 2 | 1.67 (1.36–2.07) | <0.001 | 1.72 (1.39–2.13) | <0.001 |
| 3 | 1.94 (1.57–2.39) | <0.001 | 2.15 (1.74–2.66) | <0.001 |
| >3 | 2.16 (1.74–2.67) | <0.001 | 2.26 (1.73–2.95) | <0.001 |

Abbreviation: COVID-19, Coronavirus Disease 2019; HR, hazard ratio; CI, confidence interval; SARS-CoV-2, Severe Acute Respiratory Syndrome Coronavirus-2

[a]From a multivariable Cox-proportional hazards regression model.

[b]Time-dependent covariate

[c]Calculated as number of COVID-19 nucleic acid amplification tests done per year of employment at Cleveland Clinic during the course of the pandemic.

[d]Reference: No documented prior infection.

[e]Reference: Zero doses

## Author Contributions

**Conceptualization:** Nabin K. Shrestha.

**Data curation:** Nabin K. Shrestha.

**Formal analysis:** Nabin K. Shrestha, Amy S. Nowacki.

**Investigation:** Nabin K. Shrestha, Patrick C. Burke.

**Methodology:** Nabin K. Shrestha, Amy S. Nowacki.

**Project administration:** Nabin K. Shrestha, Steven M. Gordon.

**Resources:** Patrick C. Burke, Steven M. Gordon.

**Software:** Nabin K. Shrestha.

**Supervision:** Nabin K. Shrestha.

**Validation:** Nabin K. Shrestha, Patrick C. Burke, Amy S. Nowacki.

**Visualization:** Nabin K. Shrestha, Amy S. Nowacki.

**Writing – original draft:** Nabin K. Shrestha.

**Writing – review & editing:** Nabin K. Shrestha, Patrick C. Burke, Amy S. Nowacki, Steven M. Gordon.

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
