## [Decision Letter · Decision Letter 0]

25 Aug 2023

PONE-D-23-21544Risk of Coronavirus Disease 2019 (COVID-19) among Those Up-to-Date and Not Up-to-Date on COVID-19 VaccinationPLOS ONE

Dear Dr. Shrestha,

Thank you for submitting your manuscript to PLOS ONE. After careful consideration, we feel that it has merit but does not fully meet PLOS ONE’s publication criteria as it currently stands. Therefore, we invite you to submit a revised version of the manuscript that addresses the points raised during the review process.

We look forward to receiving your revised manuscript.

Kind regards,

Cecilia Acuti Martellucci, M.D.

Academic Editor

PLOS ONE

Journal Requirements:

2. Please include captions for your Supporting Information files at the end of your manuscript, and update any in-text citations to match accordingly. Please see our Supporting Information guidelines for more information: http://journals.plos.org/plosone/s/supporting-information

Reviewers' comments:

Reviewer's Responses to Questions

**Comments to the Author**

1. Is the manuscript technically sound, and do the data support the conclusions?

Reviewer #1: Partly

Reviewer #2: Partly

Reviewer #3: Yes

2. Has the statistical analysis been performed appropriately and rigorously? 

Reviewer #1: Yes

Reviewer #2: Yes

Reviewer #3: Yes

3. Have the authors made all data underlying the findings in their manuscript fully available?

Reviewer #1: No

Reviewer #2: Yes

Reviewer #3: Yes

4. Is the manuscript presented in an intelligible fashion and written in standard English?

Reviewer #1: Yes

Reviewer #2: Yes

Reviewer #3: Yes

5. Review Comments to the Author

Reviewer #1: This clearly written paper adds to some knowledge of the efficacy of Covid-19 bivalent vaccines. However, we do have some comments that we hope will help the authors to improve the quality of their work.

Title: As PLOS ONE is aimed at an international audience, the title should indicate that "up-to-date" and "not up-to-date" refer to US-CDC definitions.

Abstract: Please provide information on the timeframe and location (Ohio, USA) of the study.

Methods:

The paper lacks information on the timing of bivalent vaccination in the "vaccinated" group. Could you add a table or figure showing the weekly or monthly distribution of this vaccination? This is important to discuss the impact of waning protection. Perhaps the time since last vaccination could be added as an explanatory variable in the model.

Please provide information on the vaccination status of the "unvaccinated group" (which is partially vaccinated, if I understand correctly).

Discussion

It's worth noting that the reference [2] does not take into account the time since the last vaccination as a confounding factor of the time period or the variant.

In my opinion, the discussion focuses too much on the results of univariate or bivariate analysis. The authors should focus on discussing the multivariate Cox model, as this model was adjusted for confounding.

“There are two reasons why not being “up-to-date” on COVID-19 vaccination by the CDC

definition was associated with a lower risk of COVID-19.” Are you referring to the raw cumulative risk or the adjusted risk of the multivariate Cox model?

The fact that the "not up-to-date" group is less at risk in the multivariate analysis, despite adjustment for covariates (in particular propensity to be tested, and prior infection) is an important result that should be discussed. What are the authors' hypotheses to explain this?

Reviewer #2: The authors examined the risk of COVID-19 in the period when the XBB variant became dominant among 48344 healthcare workers, and specifically compared the risk among those who had received at least one dose of bivalent vaccine (up to date) versus those without any bivalent vaccines (not up to date). The authors found that adults with ‘not up to date’ vaccine had a lower risk of infection than those with ‘up to date’ bivalent vaccine. I have a few concerns and comments as outlined below:

1. I noticed in the authors’ previous publication (Nabin K Shrestha et al., Effectiveness of the Coronavirus Disease 2019 Bivalent Vaccine, Open Forum Infectious Diseases, 2023), the effectiveness of bivalent vaccine was compared with non-bivalent vaccine in XBB-dominant period, and found no significant differences in risk of infection. I am wondering what the additional value of this manuscript on top of that paper is? Are there any differences in data and methodology? Also, the previous paper found no difference, while the current manuscript reported a lower risk, any explanations for this?

2. In the discussion, the authors discussed two reasons why ‘not up to date’ was associated with a lower risk of infection. First, I agree that the bivalent vaccine was less effective against XBB than BA.1 or BA.5, however, monovalent vaccine was also less effective against XBB, and studies have found an even lower neutralization of XBB by monovalent vaccine than bivalent vaccine (e.g. Kurhade, C., Zou, J., Xia, H. et al. Nat Med, 2023; Zou J, Kurhade C, Patel S, et al. N Engl J Med, 2023). Second, in the multivariable model, the authors adjusted for the most recent prior SARS-CoV-2 infection. Assuming prior infection was correctly captured, the lower adjusted HR in ‘not up to date’ group should not be affected by prior infection. Therefore, I don’t think the reasons discussed can explain the risk difference?

3. The study outcome (XBB infection) and exposure (previous infection) are sorely based on NAAT testing in employment. Little information was found regarding the testing behaviour (e.g. was it symptomatic testing? routing testing?), which can potentially induce bias.

As mentioned in the manuscript, the number of NAATs done divided by the number of years of employment was only 0.64 (0.32-1.27), which means on average each participant only had less than one test per year. This low testing frequency will likely generate a lot of missed infections. This can also be seen from Table 1, over half (53.6%) did not have documented infection before the emergence of XBB variant, which was quite high compared with similar healthcare settings, given the high infection rate in Omicron period. Therefore, missed infections are very likely to occur. While this was mentioned in the limitations, it was not thoroughly discussed and could impact the validity of the conclusion. The lower risk could be caused by the ‘not up to date’ group having more missed Omicron infections than the ‘up to date’ group, which I think is the main reason for the observed lower risk. Therefore, this evidence cannot be used to support a low effectiveness of bivalent vaccine.

4. Given the above reason, I would consider it inappropriate to conclude in the discussion and call a bivalent vaccine “questionable effectiveness”. Although bivalent vaccines have lower neutralizing capabilities against XBB variant, there is evidence showing they can provide additional protection against XBB infections (e.g. Link-Gelles R, Ciesla AA, Roper LE, et al. MMWR Morb Mortal Wkly Rep. 2023), and was associated with a lower risk of infection or severe infection with XBB variants (Lin DY, Xu Y, Gu Y, Zeng D, Sunny SK, Moore Z. N Engl J Med. 2023). These studies should be discussed. I would agree with not giving repeated boosters to all adults, but vaccines/bivalent vaccines would remain beneficial to the elderly and vulnerable groups.

5. Among 48344 participants, were all of them vaccinated? Among those ‘not up to date’ cohort, how many participants received 1, 2, 3, and 4 doses of vaccine? Would the number of vaccinations influence the results?

6. The references in the introduction and discussion are a little bit sparse, more studies on bivalent vaccine and XBB variant should be referenced and discussed, including the papers I mentioned above.

Reviewer #3: The analysis is interesting and intriquing, presentation of the results is clear and easily readable. Please add the titles to the figures presented, the titles should represent the aim and the main content of the figures.

I`m a little bit sceptic bout one sentence in the Discussion part of the manuscript, that sounds like " People may still choose to get vaccine, but an assumption that the vaccine protects against severe disease and death is not enough to unconditionally push a vaccine of questionable effectiveness to all adults". First of all, presented study does not address that issues, i.e vaccine effectiveness against severe disease, so it is quite boldly to claim it in the manuscript.

Here are some new studies about that

https://pubmed.ncbi.nlm.nih.gov/37043647/

https://pubmed.ncbi.nlm.nih.gov/37561053/

It would be polite not to do wrong conclusions about vaccine effectiveness in combating severe COVID-19 infection.

6. PLOS authors have the option to publish the peer review history of their article (what does this mean?). If published, this will include your full peer review and any attached files.

Reviewer #1: **Yes: **Vincent Auvigne

Reviewer #2: No

Reviewer #3: No

---

## [Author Response · Author response to Decision Letter 0]

13 Sep 2023

Reviewer #1: This clearly written paper adds to some knowledge of the efficacy of Covid-19 bivalent vaccines. However, we do have some comments that we hope will help the authors to improve the quality of their work.

Title: As PLOS ONE is aimed at an international audience, the title should indicate that "up-to-date" and "not up-to-date" refer to US-CDC definitions.

Response: We added this to the title.

Abstract: Please provide information on the timeframe and location (Ohio, USA) of the study.

Response: We added the location, “Ohio, USA”, in the abstract. The relevant timeframe of the study (while the COVID-19 XBB lineages were dominant) is already mentioned in the abstract.

Methods:

The paper lacks information on the timing of bivalent vaccination in the "vaccinated" group. Could you add a table or figure showing the weekly or monthly distribution of this vaccination? This is important to discuss the impact of waning protection. Perhaps the time since last vaccination could be added as an explanatory variable in the model.

Response: The paper already provides this information. The numbers at risk in Figure 1, show the weekly distribution of bivalent vaccination. Those “up-to-date” were those who had received the bivalent vaccine. Those “not up-to-date” were those who had not. We but could not add time since last vaccination as an explanatory variable, because 6174 study participants had not received a vaccine prior to the study.

Please provide information on the vaccination status of the "unvaccinated group" (which is partially vaccinated, if I understand correctly).

Response: We did not have an “unvaccinated group” in the study. The reviewer may have meant to ask what proportion of previously unvaccinated individuals received the bivalent vaccine. 37 of the 6174 previously unvaccinated individuals received the bivalent vaccine. We have added this information in the first paragraph of the results.

Discussion

It's worth noting that the reference [2] does not take into account the time since the last vaccination as a confounding factor of the time period or the variant.

Response: Reference 2 adjusts for time since proximate SARS-CoV-2 exposure, which is scientifically more sound than adjusting for time since last vaccination. The study also adjusted for the variant.

In my opinion, the discussion focuses too much on the results of univariate or bivariate analysis. The authors should focus on discussing the multivariate Cox model, as this model was adjusted for confounding.

Response: The suggestion is valid. We have removed the entire paragraph that discusses the findings of the univariable analysis. We have added a new paragraph that discusses the findings of the multivariable analysis.

“There are two reasons why not being “up-to-date” on COVID-19 vaccination by the CDC

definition was associated with a lower risk of COVID-19.” Are you referring to the raw cumulative risk or the adjusted risk of the multivariate Cox model?

Response: This was referring to the comparison of the raw cumulative risk between those “up-to-date” and those “not up-to-date”. As stated in our response to the immediate prior comment, we have removed this entire paragraph.

The fact that the "not up-to-date" group is less at risk in the multivariate analysis, despite adjustment for covariates (in particular propensity to be tested, and prior infection) is an important result that should be discussed. What are the authors' hypotheses to explain this?

Response: In response to one of the reviewers’ comments we added number of prior vaccine doses as a covariate in the multivariable model. When we did this, the association of lower risk with not being “up-to-date” was no longer apparent. We accordingly changed the study’s conclusions.

We added the following as a new paragraph to discuss the findings on multivariable analysis:

‘Specific prior infections, prior vaccinations, time since proximate exposure to SARS-CoV-2 by prior infection or vaccination, number of prior vaccine doses, and interval between vaccines, all possibly contribute to determining an individual’s risk of acquiring COVID-19. It is difficult to account for the effects of all these factors in an individual. What this study clearly shows is that a simplistic categorization of being or not being “up-to-date” on bivalent vaccination did not provide an accurate classification of risk of acquiring COVID-19 while the XBB lineages were the dominant circulating strains. A significant contributor to the lack of accuracy of classification based on receipt of this vaccine is likely to be because the vaccine is of questionable effectiveness [2].’

Reviewer #2: The authors examined the risk of COVID-19 in the period when the XBB variant became dominant among 48344 healthcare workers, and specifically compared the risk among those who had received at least one dose of bivalent vaccine (up to date) versus those without any bivalent vaccines (not up to date). The authors found that adults with ‘not up to date’ vaccine had a lower risk of infection than those with ‘up to date’ bivalent vaccine. I have a few concerns and comments as outlined below:

1. I noticed in the authors’ previous publication (Nabin K Shrestha et al., Effectiveness of the Coronavirus Disease 2019 Bivalent Vaccine, Open Forum Infectious Diseases, 2023), the effectiveness of bivalent vaccine was compared with non-bivalent vaccine in XBB-dominant period, and found no significant differences in risk of infection. I am wondering what the additional value of this manuscript on top of that paper is? Are there any differences in data and methodology? Also, the previous paper found no difference, while the current manuscript reported a lower risk, any explanations for this?

Response: The purpose of the previous paper was to examine the effectiveness of the bivalent vaccine. The primary purpose of this study was not to examine the effectiveness of the vaccine, but to examine if the CDC classification into “up-to-date” or “not up-to-date” was a useful classification system.

Our previous study only included 2 weeks of data during the XBB dominant phase of the epidemic. This study included 12 weeks of data during the XBB dominant phase of the epidemic. With the revised multivariable analysis with addition of number of prior vaccine doses as a covariate, this study also did not find a difference, so the findings are still consistent with those of the prior study.

2. In the discussion, the authors discussed two reasons why ‘not up to date’ was associated with a lower risk of infection. First, I agree that the bivalent vaccine was less effective against XBB than BA.1 or BA.5, however, monovalent vaccine was also less effective against XBB, and studies have found an even lower neutralization of XBB by monovalent vaccine than bivalent vaccine (e.g. Kurhade, C., Zou, J., Xia, H. et al. Nat Med, 2023; Zou J, Kurhade C, Patel S, et al. N Engl J Med, 2023). Second, in the multivariable model, the authors adjusted for the most recent prior SARS-CoV-2 infection. Assuming prior infection was correctly captured, the lower adjusted HR in ‘not up to date’ group should not be affected by prior infection. Therefore, I don’t think the reasons discussed can explain the risk difference?

Response: We don’t dispute that the monovalent vaccine may have also been less effective against XBB than the earlier sub-lineages. However, when this study was done, all individuals were many months away from their last monovalent vaccine, and one should not expect any residual protection from the monovalent vaccine (as many papers have shown that protection from vaccine wanes within a few months). Moreover, the definition of being “up-to-date” was not determined by whether or not a person received a monovalent vaccine, nor by how many doses of monovalent vaccine a person had received. So the assertion that monovalent vaccine was also less effective against XBB, while true, has no bearing on the question examined in this study. 

The reviewer is correct that because the multivariable model adjusted for most recent prior SARS-CoV-2 infection, prior infection should not be invoked to explain a lower HR in those “not up-to-date”. The reviewer is also correct that prior infection cannot explain a lower adjusted HR associated with the “not up to date” state, as the multivariable analysis adjusted for prior infection. The results of the revised multivariable analysis are different, and as discussed in our responses to the prior reviewer, we have removed this entire paragraph. We have added a new paragraph in its place, to discuss the results of the revised multivariable analysis.

3. The study outcome (XBB infection) and exposure (previous infection) are sorely based on NAAT testing in employment. Little information was found regarding the testing behaviour (e.g. was it symptomatic testing? routing testing?), which can potentially induce bias.

As mentioned in the manuscript, the number of NAATs done divided by the number of years of employment was only 0.64 (0.32-1.27), which means on average each participant only had less than one test per year. This low testing frequency will likely generate a lot of missed infections. This can also be seen from Table 1, over half (53.6%) did not have documented infection before the emergence of XBB variant, which was quite high compared with similar healthcare settings, given the high infection rate in Omicron period. Therefore, missed infections are very likely to occur. While this was mentioned in the limitations, it was not thoroughly discussed and could impact the validity of the conclusion. The lower risk could be caused by the ‘not up to date’ group having more missed Omicron infections than the ‘up to date’ group, which I think is the main reason for the observed lower risk. Therefore, this evidence cannot be used to support a low effectiveness of bivalent vaccine.

Response: It is not possible to retrospectively determine why individuals chose to get tested. One should not assume there would be bias by assuming that individuals “up-to-date” had different testing behavior than those “not up-to-date”. This study employed a time-dependent covariate to separate the “up-to-date” and “not up-to-date” state, so it would be the same individuals who would contribute data to the “up-to-date” state and the “not up-to-date” state. So it is not that different individuals contribute data to the different states. One could, of course, argue that getting the bivalent vaccine might have changed peoples’ testing behavior. If so, the most logical change one would expect in testing behavior in an individual is that one would be less likely to get tested for COVID-19 when “up-to-date” than when “not up-to-date”, because those who chose to receive the bivalent vaccine did so with the expectation that it would decrease their risk of COVID-19. If there was such a change in testing behavior, COVID-19 would have been less likely to have been detected in the “up-to-date” state than in the “not up-to-date” state. So it would not mask a beneficial effect of vaccination.

We agree that many infections were probably missed. By the time the later sub-lineages of the Omicron variant became dominant, COVID-19 was no longer the severe illness it was when it first appeared. At least some people would have chosen not to get tested when they had mild symptoms. However, this should not have affected the comparison between the “up-to-date” and “not up-to-date” states, because infections would have been missed in both states. As discussed in the previous section, if there was a difference in detection of infections in the two states because of testing behavior differences between the two states, it would likely be that there was comparatively less testing in the “up-to-date” state than in the “no up-to-date” state. If there was a bias in detecting COVID-19 based on difference in testing behavior, any such detection bias would have actually had the effect of masking some of the risk of COVID-19 among those “up-to-date”. We expanded the second paragraph of the discussion to explain this.

4. Given the above reason, I would consider it inappropriate to conclude in the discussion and call a bivalent vaccine “questionable effectiveness”. Although bivalent vaccines have lower neutralizing capabilities against XBB variant, there is evidence showing they can provide additional protection against XBB infections (e.g. Link-Gelles R, Ciesla AA, Roper LE, et al. MMWR Morb Mortal Wkly Rep. 2023), and was associated with a lower risk of infection or severe infection with XBB variants (Lin DY, Xu Y, Gu Y, Zeng D, Sunny SK, Moore Z. N Engl J Med. 2023). These studies should be discussed. I would agree with not giving repeated boosters to all adults, but vaccines/bivalent vaccines would remain beneficial to the elderly and vulnerable groups.

Response: We would like to clarify that this study does not draw the conclusion that the bivalent vaccine was of questionable effectiveness. This study concludes that the CDC classification into “up-to-date” and “not up-to-date” is not useful.

Our previous study failed to find the vaccine to be effective against the XBB lineages. That study was published in a major peer-reviewed infectious disease journal, and in the scores of communications we have received about the study not one provided a reasonable explanation for why that conclusion was not justified. We thus stand by the conclusion of the prior study that the vaccine was of questionable effectiveness. However, that conclusion pertains to our previous study, not to this one.

We have been aware of the two studies mentioned by the reviewer. These studies do not provide evidence that the bivalent booster provides additional protection against XBB variants. 

The first study by Link-Gelles et al, published in MMWR, examined the effectiveness of the bivalent vaccine during a time when the BA.5 and XBB lineages were dominant. During the study period, a little over a third of infections were XBB/XBB.1.5-related infections. The majority were BA.5-related infections. We have shown in one of our earlier studies that the bivalent vaccine was effective in preventing infections caused by the BA.5 lineages. In the Link-Gelles study, the lack of effectiveness of the bivalent vaccine against XBB lineages (which comprised a minority of the infections) could have easily been masked by the effectiveness of the vaccine against the BA.5 lineages (which comprised the majority of the infections). Placing this study’s findings in context of other studies on the subject, one cannot use the Link-Gelles study’s findings to claim that the bivalent vaccine protects against XBB lineage infections.

The second study by Lin et al published in the NEJM does not examine the effectiveness of the bivalent vaccine against the XBB lineages of the Omicron variant. It examines the duration of protection provided by bivalent vaccination in two cohorts, the first one consisting of individuals who received the vaccine when the BA.4/BA.5 lineages were dominant, and the second one consisting of individuals who received the vaccine when the BQ or XBB lineages were dominant. Results of the second cohort showed that the bivalent vaccine was still about 20% effective 9-10 weeks after receipt of the vaccine. It should however, be noted that the majority of the study occurred before the XBB lineages became dominant. Only a very small proportion of the study occurred when the XBB lineages were dominant, and the XBB lineages only became dominant during the last two weeks of the study. There are data only for 2-3 weeks for those who received the bivalent vaccine during XBB lineage dominance, and lack of effectiveness against the XBB lineages could easily have been masked by effectiveness against the BQ lineages, as most of the data for the first three weeks since receipt of the bivalent booster would have consisted of data accrued while the BQ lineages were dominant. This study in no way shows that the bivalent vaccine protected against the XBB lineages of the Omicron variant.

There have been no studies that have actually shown that the bivalent vaccine is beneficial in the elderly or in the immunocompromised. It is widely understood that the elderly and the immunocompromised have weaker immune responses than an otherwise younger and healthier population. No vaccine has ever been shown to be beneficial in the immunocompromised when it is not beneficial in those who are not immunocompromised. The reason for strongly encouraging vaccination of the elderly and immunocompromised is not because they are expected to have a stronger immune response, but because they are at higher risk of adverse consequences of the infection and any protection, even if weaker than in a healthier population, is worth the effort of vaccination. But this, of course, assumes that the vaccine is effective among those able to mount an immune response. There is no logic to the assertion that a vaccine that is not effective in healthier individuals would miraculously be effective among the immunocompromised.

5. Among 48344 participants, were all of them vaccinated? Among those ‘not up to date’ cohort, how many participants received 1, 2, 3, and 4 doses of vaccine? Would the number of vaccinations influence the results?

Response: Of the 48344 participants, 6174 were not vaccinated at the start of the study. In table 1, we have added a breakdown of the number of vaccine doses received by the study participants at the start of the study. Adding the number of prior vaccine doses did indeed influence the results greatly! We have included this as a variable in the multivariable model. As we have found in prior studies, there was an association of increased risk of COVID-19 with higher number of prior vaccine doses even after adjustment for other factors. However, once this was added to the model, there was no longer an association of higher risk of COVID-19 with being “up-to-date” with the bivalent vaccine. Thank you for raising this question. It made a huge difference in the results. 

Given the change in the results, we changed the reference level of the bivalent vaccinated status from “up-to-date” to “not up-to-date”, so that the results could be worded more simply. We also accordingly revised our conclusion to say that being “up-to-date” on COVID-19 vaccination by the CDC definition was not associated with a lower risk of COVID-19 than not being “up-to-date”.

6. The references in the introduction and discussion are a little bit sparse, more studies on bivalent vaccine and XBB variant should be referenced and discussed, including the papers I mentioned above.

Response: We acknowledge the suggestion to add additional references. Which references to include is a matter of writing style preference. We believe that we have included the references that support statements we have made in the introduction and discussion. We acknowledge that there are many other papers on the subject, but we did not think we have excluded any that would substantially alter the rationale for doing the study or placing the findings into context. Specifically, we chose not to reference the mentioned papers because their findings are either tangential to the issues being discussed or their claims are not supported by the data presented. Had we been able to find studies that found the bivalent vaccine to be effective against the XBB lineages, we would have certainly referenced them.

Reviewer #3: The analysis is interesting and intriquing, presentation of the results is clear and easily readable. Please add the titles to the figures presented, the titles should represent the aim and the main content of the figures.

Response: The figures have the appropriate titles. They may not have shown up properly in the version sent out for peer review. We understand that the figures will have the appropriate titles in the final printed version. 

I`m a little bit sceptic bout one sentence in the Discussion part of the manuscript, that sounds like " People may still choose to get vaccine, but an assumption that the vaccine protects against severe disease and death is not enough to unconditionally push a vaccine of questionable effectiveness to all adults". First of all, presented study does not address that issues, i.e vaccine effectiveness against severe disease, so it is quite boldly to claim it in the manuscript.

Here are some new studies about that

https://pubmed.ncbi.nlm.nih.gov/37043647/

https://pubmed.ncbi.nlm.nih.gov/37561053/

It would be polite not to do wrong conclusions about vaccine effectiveness in combating severe COVID-19 infection.

Response:

We think that statement is very appropriate. In a prior study we failed to find the bivalent vaccine to be effective in preventing infection caused by the XBB lineages of the Omicron variant. We have not come across a single study that has truly shown the bivalent vaccine provided protection against severe infection caused by the XBB lineages. Moreover, in this and our prior study we found an association of increased risk of COVID-19 with increasing number of prior vaccine doses, in multivariable analyses. We have not said that people should not get the vaccine. We make the argument that one should not mandate a vaccine whose effectiveness is questionable.

In the next two paragraphs we specifically address the two papers suggested to be included as references, to explain why they were not included as references:

We have already described why the first paper (Lin et al NEJM 2023) does not make the case for bivalent vaccine effectiveness, in our response to comment # 4 from reviewer #2.

The findings as presented for the second paper also do not support the study’s assertion that the bivalent vaccine protected against COVID-19. Looking at table 2, which provides the main results of the study, a person who is paying attention will notice that the incidence rate per 100,000 was 0.587 in the reference group and 0.586 for the 15-60 day group. The incidence rate is almost identical in those groups. How does that square with the calculated relative vaccine effectiveness being 45.6%? There are similar questions for each of the groups. Perusal of the presented data shows that the incidence of severe COVID-19 was similar for each group (different time intervals since receipt of second bivalent booster) compared to the reference group (those who received only one booster dose), suggesting that receiving more than one booster dose provided no additional protection over receipt of only one booster dose. Even if we are misreading the results and there are no flaws in the methodology, this study was a comparison between the effectiveness of one booster dose versus multiple booster doses, not an evaluation of the effectiveness of the bivalent vaccine.

We agree that it is not helpful to draw wrong conclusions about vaccine effectiveness against severe COVID-19. This study does not make a conclusion about vaccine effectiveness against severe COVID-19. It only makes a conclusion about the usefulness of the “up-to-date” / “not up-to-date” classification.

---

## [Decision Letter · Decision Letter 1]

13 Oct 2023

Risk of Coronavirus Disease 2019 (COVID-19) among Those Up-to-Date and Not Up-to-Date on COVID-19 Vaccination by US CDC Criteria

PONE-D-23-21544R1

Dear Dr. Shrestha,

We’re pleased to inform you that your manuscript has been judged scientifically suitable for publication and will be formally accepted for publication once it meets all outstanding technical requirements.

Kind regards,

Cecilia Acuti Martellucci, M.D.

Academic Editor

PLOS ONE

Additional Editor Comments (optional):

Reviewers' comments:

Reviewer's Responses to Questions

**Comments to the Author**

1. If the authors have adequately addressed your comments raised in a previous round of review and you feel that this manuscript is now acceptable for publication, you may indicate that here to bypass the “Comments to the Author” section, enter your conflict of interest statement in the “Confidential to Editor” section, and submit your "Accept" recommendation.

Reviewer #1: (No Response)

Reviewer #2: All comments have been addressed

Reviewer #3: All comments have been addressed

2. Is the manuscript technically sound, and do the data support the conclusions?

Reviewer #1: Yes

Reviewer #2: Yes

Reviewer #3: Yes

3. Has the statistical analysis been performed appropriately and rigorously? 

Reviewer #1: Yes

Reviewer #2: Yes

Reviewer #3: Yes

4. Have the authors made all data underlying the findings in their manuscript fully available?

Reviewer #1: Yes

Reviewer #2: Yes

Reviewer #3: Yes

5. Is the manuscript presented in an intelligible fashion and written in standard English?

Reviewer #1: Yes

Reviewer #2: Yes

Reviewer #3: Yes

6. Review Comments to the Author

Reviewer #1: Most of our comments have been taken into account by the authors. The article is now well focused on the evaluation of the US CDC vaccination criteria on the risk of Covid-19, which was its main objective.

Reviewer #2: The authors have addressed all my comments and I do not have any further comments. I believe the manuscript is much improved and the messages are clearly presented.

Reviewer #3: No comments here. Thanks for exciting research and staying committed to your idea. Your research adds freshness to the science and exactly enhance negotiations about vaccination plan implementation.

7. PLOS authors have the option to publish the peer review history of their article (what does this mean?). If published, this will include your full peer review and any attached files.

Reviewer #1: **Yes: **Vincent Auvigne

Reviewer #2: No

Reviewer #3: No

---

## [Editor Report · Acceptance letter]

17 Oct 2023

PONE-D-23-21544R1 

Risk of Coronavirus Disease 2019 (COVID-19) among Those Up-to-Date and Not Up-to-Date on COVID-19 Vaccination by US CDC Criteria 

Dear Dr. Shrestha:

I'm pleased to inform you that your manuscript has been deemed suitable for publication in PLOS ONE. Congratulations! Your manuscript is now with our production department. 

Kind regards, 

on behalf of

Dr. Cecilia Acuti Martellucci 

Academic Editor

PLOS ONE